# Microglial CD74 Expression Is Regulated by TGFβ Signaling

**DOI:** 10.3390/ijms231810247

**Published:** 2022-09-06

**Authors:** Jannik Jahn, Antonia Bollensdorf, Christopher Kalischer, Robin Piecha, Jana Weiß-Müller, Phani Sankar Potru, Tamara Ruß, Björn Spittau

**Affiliations:** 1Institute of Anatomy, University Medicine Rostock, University of Rostock, 18051 Rostock, Germany; 2Anatomy and Cell Biology, Medical School OWL, Bielefeld University, 33615 Bielefeld, Germany

**Keywords:** CD74, microglia, TGFβ1, LPS

## Abstract

**Simple Summary:**

In the present study, we provided evidence that TGFβ signaling regulated the expression of the microglia activation marker CD74. Our data demonstrated that TGFβ1 inhibited LPS-induced upregulation of CD74. Moreover, inhibition of microglial TGFβ signaling in vitro and silencing of TGFβ signaling by deletion of Tgfbr2 in vivo resulted in marked upregulation of microglial CD74.

**Abstract:**

Microglia play important roles during physiological and pathological situations in the CNS. Several reports have described the expression of *Cd74* in disease-associated and aged microglia. Here, we demonstrated that TGFβ1 controled the expression of *Cd74* in microglia in vitro and in vivo. Using BV2 cells, primary microglia cultures as well as *Cx3cr1^CreERT2^:R26-YFP:Tgfbr2^fl/fl^* in combination with qPCR, flow cytometry, and immunohistochemistry, we were able to provide evidence that TGFβ1 inhibited LPS-induced upregulation of *Cd74* in microglia. Interestingly, TGFβ1 alone was able to mediate downregulation of CD74 in vitro. Moreover, silencing of TGFβ signaling in vivo resulted in marked upregulation of CD74, further underlining the importance of microglial TGFβ signaling during regulation of microglia activation. Taken together, our data indicated that CD74 is a marker for activated microglia and further demonstrated that microglial TGFβ signaling is important for regulation of *Cd74* expression during microglia activation.

## 1. Introduction

As the resident immune cells of the central nervous system (CNS), microglia are involved in a plethora of pathological as well as physiological processes [1,2]. Microglia constantly scan their local microenvironment sensing impairments triggered by endogenous and/or exogenous factors [3]. Damage-associated molecular patterns (DAMPs) and pathogen-associated molecular patterns (PAMPs) have been described and are recognized by Toll-like receptors (TLRs) or NOD-like receptors (NLRs), subsequently inducing microglia activation [4]. Recent sophisticated in vivo studies using (single cell) RNA-sequencing have defined a highly conserved transcriptional profile under neurodegenerative conditions including upregulation of *ApoE*, *Axl*, *Clec7a*, *Cst7*, *Cybb*, *Ctsd*, *Il1b*, *Itgax*, *Lgals*, *Lilrb4*, *Lpl*, *Nos2*, *Spp1*, *Trem2*, as well as *Tyrobp* [5,6,7]. These data lead to the concept that microglia exist as homeostatic cells under basal conditions and adopt an activation profile under pathological conditions that defines them as disease-associated microglia (DAM). Noteworthy, DAMs can also shift toward a neurodegenerative microglia phenotype (MGnDs), depending on the nature of the neuropathology and the severity of the disease [8]. Interestingly, recent reports have described that DAMs as well as aged microglia display a partially overlapping transcriptional profile which also includes increased expression of *Cd74* [7,9,10].

Transforming growth factor β1 (TGFβ1) has been shown to be an essential endogenous cue for postnatal microglia maturation by induction of a unique transcriptional signature including upregulation of genes such as as *Transmembrane protein 119* (*Tmem119*), *Purinergic Receptor P2Y12* (*P2ry12*), *Olfactomedin-like 3* (*Olf*mL*3*), *Sal-like 1* (*Sall1*), *G protein receptor 34* (*Gpr34*), *Hexosa*min*idase Beta* (*Hexb*), and *Fc receptor-like S* (*Fcrls*) [5,11,12]. Moreover, TGFβ1 is able to abrogate microglia activation induced by lipopolysaccharide (LPS) and interferon-γ (IFNγ) in vitro [13,14]. Silencing of microglial TGFβ signaling or lack of extracellular TGFβ binding and processing in vivo results in loss of microglia maturation, and enhanced microglia activation has been demonstrated underlining the importance of TGFβ1 as a potent immunosuppressive factor for microglia [12,15,16].

CD74 is a MHC class II-associated invariant chain and regulates trafficking of MHCII molecules in antigen-presenting cells [17]. However, CD74 has been demonstrated to be involved in several signaling pathways. For instance, CD74 serves as a receptor for the cytokine macrophage migration inhibitory factor (MIF), which regulates a broad range of immune cell function and inflammatory reactions [18,19,20]. Increased expression levels of *Cd74* have been reported in activated microglia in different neuropathological models and, thus, is considered as a microglia activation marker [21,22,23].

In the current study, the effects of TGFβ1 as well as microglial TGFβ signaling on *Cd74* expression were analyzed in vitro using BV2 cells, primary microglia, and mixed glia cultures. Moreover, transgenic mice with microglia-specific deletion of *Tgfbr2* were employed to demonstrate that TGFβ signaling inhibited expression of *Cd74* under basal conditions and further abrogated LPS-induced transcriptional activation of *Cd74*. Together, these data further confirmed the essential immunosuppressive roles of TGFβ1 for microglia and clearly demonstrated that TGFβ regulated expression of *Cd74* in vitro and in vivo.

## 2. Results

### 2.1. TGFβ1 Inhibits LPS-Mediated Upregulation of Cd74 in BV2 Cells

Using RNA sequencing, we have recently demonstrated that *Cd74* was upregulated in microglia isolated from adult transgenic mice with deficient TGFβ signaling [18]. In order to validate whether expression of microglial *Cd74* is directly regulated by TGFβ1, we first used the microglia cell line BV2. Cells were treated with TGFβ1 (5 ng/mL), LPS (1 µg/mL), or a combination of both factors for 6 h, 12 h, and 24 h under serum-free conditions. As depicted in Figure 1A–C, treatment with TGFβ1 alone resulted in significant downregulation of *Cd74* expression, whereas treatment with LPS induced upregulation of *Cd74* mRNA levels at all time points analyzed. Interestingly, TGFβ1 blocked the LPS-induced increase of *Cd74* transcripts at 6 h (Figure 1A), 12 h (Figure 1B), and 24 h (Figure 1C). Analysis of CD74 protein levels after treatment of BV2 cells confirmed the transcriptional data. Again, TGFβ1 treatment alone resulted in significantly reduced CD74 total protein levels compared to control cells after 6 h (Figure 1D,G), 12 h (Figure 1E,H), and 24 h (Figure 1F,I). Treatments with LPS resulted in significant increases of CD74 protein levels at all time points, reaching a maximum at 12 h (Figure 1D–I). In accordance with the abovementioned qPCR data, TGFβ1 significantly inhibited the LPS-driven increases in CD74 total protein levels in BV2 cells at all experimental time points (Figure 1D–I). Together, these data demonstrated that TGFβ1 downregulated expression of *Cd74* and further blocked the LPS-mediated upregulation of *Cd74* in BV2 cells.

### 2.2. TGFβ1 Inhibits LPS-Mediated Upregulation of Cd74 in Primary Mouse Microglia

In order to confirm the data obtained from BV2 cells, primary mouse microglia were used during the next experimental steps. Since the availability of primary microglia cells is limited, we decided to exclusively use 6 h and 24 h as experimental time points with this approach. Figure 2 shows that treatment of primary microglia with TGFβ1 resulted in significant downregulation of microglial *Cd74* expression after 6 h (Figure 2A) and 24 h (Figure 2B). Comparable with the results from BV2 cells, TGFβ1 significantly abrogated the LPS-induced upregulation of *Cd74* expression in primary microglia (Figure 2A,B). As depicted in Figure 2C,E, TGFβ1 did not affect CD74 total protein levels in primary microglia after 6 h, whereas LPS induced increased levels of CD74 without reaching significancy. After treatment for 24 h, TGFβ1 significantly decreased CD74 total protein levels and efficiently inhibited LPS-induced increases of CD74 in primary microglia (Figure 2D,F). The densitometric analysis of the distinct CD74 31 kDa and 41 kDa isoform levels are depicted in Appendix A. These data confirmed the results obtained in the microglia cell line BV2 and clearly demonstrated that TGFβ1 alone decreased *Cd74* expression and sufficiently inhibited LPS-triggered upregulation of *Cd74* in primary microglia cultures.

### 2.3. Inhibition of TGFβ Signaling Increases CD74 Cell Surface Levels in Primary Mouse Microglia

In the next step, blocking of TGFβ signaling using a TGFβ receptor type I inhibitor (TβRI) was employed to analyze the effects of microglial TGFβ signaling in mixed glia cultures. As a prerequisite, the functionality of the inhibitor was assessed after TGFβ1 treatment of primary microglia and subsequent analysis of SMAD2/3 nuclear translocation. Figure 3 demonstrates that treatment of microglia with TGFβ1 for 2 h resulted in nuclear accumulation of SMADs whereas co-treatment with the TβRI virtually completely blocked TGFβ1-induced shuttling of SMADs to the nuclei (Figure 3A,B).

In contrast to the abovementioned experimental approach, where primary microglia cultures were used under serum-free conditions, experiments involving pharmacological inhibition of TGFβ signaling were performed using mixed glia cultures containing 10% serum (FCS) during all steps. Initially, mature mixed glia/microglia cultures were treated with TβRI or LPS for 24 h, microglia were shaken off, and RNA as well as protein isolation were performed for subsequent qPCR and western blotting (Figure 4A).

As shown in Figure 4B, inhibition of TGFβ signaling resulted in increased transcription of *Cd74*, whereas LPS treatment significantly reduced *Cd74* RNA levels after 24 h. As expected, total CD74 protein levels were significantly increased after treatment with TβRI (Figure 4C,D). Interestingly, LPS significantly increased the total CD74 protein levels after 24 h treatment, which was in contrast to the observed qPCR data. Noteworthy, LPS enhanced the abundancy of the 41 kDa variant of CD74, whereas inhibition of TGFβ signaling exclusively increased the 31 kDa variant of CD74 (Figure 4C,D). The densitometric analysis of 31 kDa and 41 kDa protein levels after treatment with TβRI or LPS for 24 h is depicted in Appendix A.

In the next step, we analyzed the surface expression of CD74 in primary microglia cultures using flow cytometry. As depicted in Figure 5A, mixed glia cultures were treated with a TGFβ receptor inhibitor (5 nM TβRI) or lipopolysaccharide (LPS, 1 µg/mL) for 1, 3, 5, and 7 days, and microglia were finally shaken off and used for flow cytometry. Figure 5B shows the gating strategy to identify CD74^+^ microglia out of the F4/80^+^ population. The initial flow cytometry analysis revealed that most of the F4/80^+^ microglia were also positive for CD74 (Figure 5B). Thus, we decided to further analyze the changes in relative fluorescence intensities after different treatments and time points. We observed that inhibition of TGFβ signaling in mixed glia cultures resulted in increased surface levels of CD74 in microglia at all time points analyzed (Figure 5C–F). Interestingly, the surface levels of CD74 after LPS treatment were significantly reduced compared to those of untreated control cultures (Figure 5C–F). These data indicated that TGFβ signaling was vital to control CD74 surface levels in microglia and that the LPS-induced upregulation might be abrogated by inhibitory factors and/or molecules released by cells in the mixed glia culture setup.

### 2.4. Microglia-Specific Knockout of Tgfbr2 Increases Microglial CD74 and MHCII Expression In Vivo

To further elucidate the importance of TGFβ signaling for the regulation of *Cd74* expression in vivo, we used transgenic mice with microglia-specific deletion of *Tgfbr2* [18]. As shown in Figure 6A, new born *Cx3cr1^CreERT2^:R26-YFP:Tgfbr2^fl/fl^* and control mice received tamoxifen injections (0.2 mg/10 µL) on postnatal days 3 and 5. Subsequently, brains were isolated and fixed at postnatal day 14 and used for immunohistochemistry. Whereas mice lacking expression of Cre recombinase showed no YFP^+^ microglia (Figure 6B,H), the vast majority (98.99% ± 0.4978%) of microglia (Iba1^+^) in brain sections from *Cx3cr1^CreERT2^:R26-YFP:Tgfbr2^fl/fl^* mice displayed positivity for YFP after tamoxifen-induced recombination (Figure 6C,H), indicating sufficient deletion of exons 2 and 3 of *Tgfbr2* in microglia. Next, the expression of CD74 in microglia of control and *Cx3cr1^CreERT2^:R26-YFP:Tgfbr2^fl/fl^* mice was analyzed. As shown Figure 6D,I, 3.928% (±0.3991%) of microglia in control mice were positive for Iba1 and CD74. In contrast, virtually all microglia (99.18% ± 0.1019%) in *Tgfbr2*-deficient mice showed strong positivity for CD74 (Figure 6E,I). The strong functional link between CD74 and MHC class II led us to analyze the expression of MHCII in *Cx3cr1^CreERT2^:R26-YFP:Tgfbr2^fl/fl^* mice. Similar to the low expression of CD74, control microglia showed only a very weak (14.67% ± 2.553%) immunoreactivity for MHCII (Figure 6F,J). Noteworthy, almost all Iba1^+^ microglia in *Cx3cr1^CreERT2^:R26-YFP:Tgfbr2^fl/fl^* mice showed robust expression of MHCII (99.08% ± 0.09112%) after tamoxifen-induced recombination (Figure 6G,J). Together, these data confirmed the observations made during in vitro experiments and clearly demonstrated that silencing of microglial TGFβ signaling in vivo, by knockout of exons 2 and 3 of *Tgfbr2*, resulted in upregulation of CD74 and MHCII expression, indicating microglia activation and the increase in antigen presentation.

## 3. Discussion

In the present study, we demonstrated that TGFβ1 inhibited LPS-induced upregulation of *Cd74* in the microglia cell line BV2 as well as in primary mouse microglia. Moreover, treatment with TGFβ1 alone resulted in significant downregulation of *Cd74* in both cell culture systems. Interestingly, we provided evidence that LPS predominantly increased the 41 kDa chain and inhibition of TGFβ signaling resulted in upregulation of the 31 kDa isoform. Using flow cytometry, we clearly showed that inhibition of TGFβ signaling in mixed glial cultures increased the surface levels of CD74 in vitro. Finally, silencing of microglial TGFβ signaling in vivo using *Cx3cr1^CreERT2^:R26-YFP:Tgfbr2^fl/fl^* mice confirmed our in vitro data. Here, lack of *Tgfbr2* in microglia resulted in robust microglia activation associated with intense CD74 and MHCII immunoreactivity. Together, these data validated CD74 as a microglia activation marker and further underlined the importance of TGFβ1 and microglial TGFβ signaling to regulate expression of *Cd74* and, thus, control microglia activation.

The analysis of total proteins from BV2 cells and primary microglia in vitro revealed that CD74 was present in the 31 kDa (p31) and 41 kDa (p41) forms. Interestingly, LPS treatment increased p41, whereas inhibition of TGFβ signaling increased the p31 form of CD74. Moreover, the increase in the 41 kDa isoform triggered by LPS treatment was faster in BV2 cells compared to primary microglia, which might be caused by the well-described activated phenotype of BV2 cells. These two isoforms existed due to alternative splicing [24], and the observed expression suggested that LPS and inhibition of TGFβ signaling activate different pathways, resulting in differences in CD74 isoform expression. Studies involving transgenic mice with exclusive expression of specific CD74 isoforms have demonstrated that in general all isoforms are able to mediate MHCII assembly, transport, and subsequent antigen presentation [24,25]. However, subtle differences of presented peptides have been described [26]. To which extent the observed differences in CD74 isoform expression in microglia contribute to functional aspects remains unclear. Absence of the p41 CD74 isoform seems to prevent associations between CD74 and CD44 in human lung adenocarcinoma-derived cells [27]. Moreover, the invariant chain p41 mediates production of soluble MHC class II molecules [28]. Noteworthy, an inhibitory fragment from the p41 isoform of CD74 has been shown to regulate activity of cysteine cathepsins in antigen presentation, suggesting that regulation of the proteolytic activity of most of the cysteine cathepsins by p41 is an important control mechanism of antigen presentation [29]. Inhibition of cathepsin L was shown to alleviate microglia-mediated neuroinflammatory responses through caspase-8 and NF-κB pathways [30]. It remains to be elucidated why inhibition of TGFβ signaling exclusively increased the levels of the 31 kDa isoform of CD74. As long as the function of this isoform in microglia is not understood, we can only speculate about the functional outcome of the CD74 31 kDa upregulation. However, all recent data using inhibition of microglia TGFβ signaling resulted in an activation phenotype of microglia [18,31].

Another very exciting feature of CD74 is that MIF binding induces its intramembrane cleavage and the release of the cytosolic intracellular domain (CD74-ICD). CD74-ICD is able to interact with the transcription factors Runt-related transcription factor (RUNX) and NF-κB and binds to regulatory sites of genes involved in apoptosis, immune response, and immune cell migration [32]. Based on these reports, CD74 might have anti-inflammatory as well as inflammatory functions.

Using *Cx3cr1^CreERT2^:R26-YFP:Tgfbr2^fl/fl^* mice, we clearly demonstrated that lack microglial TGFβ signaling resulted in intense CD74 immunoreactivity of microglia indicating microglia activation. It is worth to speculate about the functions of CD74 in this context. It has been demonstrated that MIF induces astrocyte activation by binding to CD74 [33]. Interestingly, the targeted inhibition of CD74 attenuates adipose COX-2-MIF-mediated M1 macrophage polarization and abrogates obesity-related adipose tissue inflammation and insulin resistance [34]. Several studies have linked high expression of CD74 with increased infiltration of immune cells in different diseases and disease models. For instance, increased infiltration of immune cells in breast cancer has been described after upregulation of CD74 [30]. Moreover, traumatic brain injury causes selective, CD74-dependent peripheral lymphocyte activation that exacerbates neurodegeneration [31]. Using MIF- and CD74-deficient mice, it was shown that CCL2-induced leukocyte adhesion and transmigration are dependent on MIF and CD74 [35]. Taken together, TGFβ-mediated downregulation of CD74 in vivo might be essential to control excessive microglia activation. Given the fact that predominantly neurons seem to express and release TGFβ1 in vivo [36], TGFβ1 should be considered as an essential factor in neuron-microglia-crosstalk under physiological and pathological conditions.

In conclusion, this study provides evidence that microglia activation was associated with increased expression of CD74 and that microglial expression of CD74 was tightly controlled by TGFβ signaling in vitro and in vivo. It remains unclear whether TGFβ directly controls CD74 expression or whether crosstalk with other signaling pathways is involved in this regulation. Moreover, the functional aspects of microglial CD74 in distinct activation settings need to be addressed in future studies using microglia-specific targeting of *Cd74*.

## 4. Materials and Methods

### 4.1. Animals

Throughout the study, NMRI mice were used for the establishment of primary microglia cultures. The generation of microglia-specific *Tgfbr2*-knockout mice has been described elsewhere [18]. Briefly, the mouse lines *Cx3cr1^CreERT2^* [37], *B6.129 × 1-Gt(ROSA)26Sortm1(EYFP)Cos/J* [38], and *Tgfbr2^flox/flox^* [39] were used to obtain *Cx3cr1^CreERT2^:R26-YFP:Tgfbr2^fl/fl^* mice. Cre recombinase activation was induced by injection of 0.2 mg tamoxifen (TAM, T5648; Sigma-Aldrich, Schnelldorf, Germany) and solved in 10 µL corn oil (C8367, Sigma-Aldrich) at postnatal days 3 (P3) and 5 (P5). Mice carrying one allele Cre (Cre/+) were considered as knockouts, whereas mice lacking Cre expression (+/+) were referred to as control mice.

All mice were obtained from Janvier (Le Genest-Saint-Isle, France) and were kept at 22 ± 2 °C under a 12 h light/dark cycle with *ad libitum* access to water and chow. All mice procedures were performed in accordance with the German Federal Animal Welfare Law and the local ethical guidelines of the University of Rostock. Experiments involving mice were approved by the animal experimentation committee of the University of Rostock and the Landesamt für Landwirtschaft, Lebensmittelsicherheit und Fischerei Mecklenburg-Vorpommern (7221.3-1-064/18).

### 4.2. Reagents

Primary microglia cultures as well as BV2 cells were treated with the following factors and reagents: 5 ng/mL TGFβ1 (Peprotech, Hamburg, Germany), 1 µg/mL LPS (Sigma-Aldrich, Schnelldorf, Germany), and 500 nM TGFβ receptor type I inhibitor (TβRI, Calbiochem, Merck, Germany).

### 4.3. BV2 Cell Culture

The microglia cell line BV2 was kept in DMEM/F12 media (Thermo Fisher Scientific, Dreieich, Germany) supplemented with 10% heat-inactivated FCS (PAN Biotech) and 1% penicillin/streptomycin (Sigma-Aldrich, Schnelldorf, Germany). BV2 cells were incubated at 37 °C in a 5% CO_2_ and 95% humidified atmosphere. Cells were washed with PBS and kept under serum-free conditions for at least 2 h prior to treatment with TGFβ1 and/or LPS for 6 h, 12 h, and 24 h.

### 4.4. Primary Microglia Cultures

Primary microglia cultures were established as described previously [31]. Briefly, brains from new born P0/P1 NMRI mice were washed with Hank’s balanced salt solutions (HBSS, Gibco), and all blood vessels and meninges were rapidly removed. Next, brains were transferred to ice-cold HBSS and digested with 1× Trypsin-EDTA (Thermo Fisher Scientific, Dreieich, Germany) at 37 °C for 10 min. An equal amount of ice-cold fetal calf serum (FCS) containing DNase (Roche, Mannheim, Germany) at a final concentration of 0.5 mg/mL was added, and finally, brains were dissociated using Pasteur pipettes. Dissociated cells were centrifuged, collected and resuspended in DMEM/F12-culturing media containing 10% FCS and 1% penicillin/streptomycin (Sigma-Aldrich, Schnelldorf, Germany). Dissociated cells from 2–3 brains were transferred into poly-D-lysine-coated (Sigma-Aldrich, Schnelldorf, Germany) 75 cm^2^ tissue culture flasks. For 25 cm^2^ flasks, cells from 1 P0/P1 brain were used.

### 4.5. RNA Isolation, Reverse Transcription, and Quantitative RT-PCR

Total RNA was isolated from primary microglia or BV2 cells using TRIfast (VWR, Darmstadt, Germany) according to the manufacturer’s instructions. RNA concentrations were analyzed using a photometer (Eppendorf BioPhotometer D30). cDNA synthesis was performed using a ProtoScript II First Strand cDNA Synthesis Kit (New England BioLabs) according to the manufacturer’s instructions. A CFX Connect^TM^ System (Bio-Rad, München, Germany) was employed for quantitative RT-PCR (qPCR) analyses. Samples were prepared using a Luna Universal qPCR Master Mix (New England BioLabs, Ipswich, MA, USA). All qPCR reactions were performed in duplicates, and the results were analyzed using the CFX Connect^TM^ System software and the comparative CT method. Data are presented as 2^−∆∆CT^ for the gene of interest (*Cd74*) normalized to the housekeeping gene Gapdh and presented as percent of the control groups. The following primers were used: *Cd74* for 5′-CCGCCTAGACAAGCTGACC-3′, *Cd74* rev 5′-ACAGGTTTGGCAGATTTCGGA-3′ (NM_010545.3), Gapdhfor 5′-AGGTCGGTGTGAACGGATTTG-3′, and Gapdhrev 5′-TGTAGACCATGTAGTTGAGGTCA-3′ (NM_008084).

### 4.6. Protein Isolation and Western Blotting

Total proteins were isolated from microglia cultures using Pierce^TM^ RIPA buffer (Thermo Fisher Scientific, Dreieich, Germany), and protein concentrations were measured using a Pierce^TM^ BCA Protein Assay Kit (Thermo Fisher Scientific, Dreieich, Germany) according to the manufacturer’s instructions. Lysates (10 µg total protein/lane) were loaded on Mini-PROTEAN Precast gels (Bio-Rad, München, Germany) for electrophoresis. Blotting was performed using a Trans-Blot^R^ Turbo^TM^ Transfer System and a Trans-Blot^R^ Turbo^TM^ RTA Midi PVDF Transfer Kit (Bio-Rad, München, Germany). All membranes were blocked with 5% BSA (Sigma-Aldrich) in TBST for 90 min. Incubation with primary antibodies against CD74 (BD Biosciences, 555317) and β-Actin (Cell Signaling Technologies, 4970) was performed at 4 °C overnight. Finally, membranes were washed with TBST and incubated with HRP-conjugated anti-rat (Abcam, ab97057) and anti-rabbit (Cell Signaling Technologies, 7074S) secondary antibodies. Labelled proteins were detected using a Pierce^TM^ ECL Western Blotting Substrate (Thermo Fisher Scientific, Dreieich, Germany). Blots were captured using a Proxima ECL Detection Setup (Isogen). Densitometric analysis was performed using ImageJ software (National Institutes of Health, Bethesda, MD, USA).

### 4.7. Immunocytochemistry and Immunohistochemistry

Primary microglia were shaken off from mixed glial cultures, plated on coverslips and incubated at 37 °C prior to treatments. After treatments, cells were washed with PBS and fixed with 4% paraformaldehyde (PFA) for 15 min, and washing with PBS (3 *×* 5 min) was performed prior to blocking with PBS containing 10% normal goat sera and 0.1% Triton-X 100 (Sigma-Aldrich, Germany) for 1 h. Microglia were incubated with anti-Smad1/2/3 (sc-7960, Santa Cruz Biotechnology Inc., Dallas, TX, USA) at 4 °C overnight. After washing three times with PBS, cells were incubated with Alexa Fluor-594-conjugated secondary antibodies (1:500, Abcam, ab150080) for 1 h. FITC-coupled isolectin (Invitrogen, Waltham, MA, USA, 121411) served as a microglial marker, and nuclei were stained using 4′-6′-diamidino-2-phenylindole (DAPI, Dianova, 711-165-152). Finally, coverslips were mounted on objective slides using Fluoromount-G mounting media (SouthernBiotech).

For immunohistochemistry, new born *Cx3cr1^CreERT2^:R26-YFP:Tgfbr2^fl/fl^* and control mice received tamoxifen injections (0.2 mg/10 µL) at postnatal days 3 and 5. Subsequently, brains were isolated and fixed at postnatal day 14, and 50 µm cryostat sections were stained using anti-Iba1 (Wako Chemicals, 019-19741), anti-GFP (Abcam, ab13970), anti-CD74 (BD Biosciences, 55317), and anti-MHC-II (Santa Cruz, sc-59318) primary antibodies. Alexa Fluor^R^ 488 anti-chicken (Abcam, ab150173), Alexa Fluor^R^ 488 anti-rat (Abcam, ab150157), and Alexa Fluor^R^ 594 anti-rabbit (Abcam, ab150080) secondary antibodies were used. DAPI was used as a nuclear counterstain, and sections were mounted using Fluoromount-G mounting media (SouthernBiotech, Birmingham, AL, USA). All fluorescence images were taken using the Nikon C1 confocal microscope (Nikon, Düsseldorf, Germany). Quantifications of cells was performed using ImageJ software (National Institutes of Health, Bethesda, MD, USA).

### 4.8. Flow Cytometry

Primary microglia were shaken off from mixed glia cultures after treatments and were stained with primary antibodies directed against F4/80 (4 µL, MCA497A488, AbD Serotech) and CD74 (3 µL, 151004, BioLegend, San Diego, CA, USA) at 4 °C for 30 min. Fc receptor blocking was performed for all samples using TrueStain fcX (101319, Biolegend) to avoid unspecific antibody binding. Finally, cells were washed with 500 µL FACS buffer, centrifuged for 5 min at 400× *g* and analyzed using a CytoFlex cytometer (Beckman Coulter, Brea, CA, USA). Quantifications were evaluated using the FlowJo^TM^ v10.8 software (BD Life Sciences, Aalst, Balgium).

### 4.9. Statistics

All data presented here are given as means ± SEM. Multiple group analysis was conducted using one-way ANOVA followed by Tukey’s multiple comparison test. *p*-values < 0.05 were considered as being statistically significant. The software GraphPad Prism 8 (GraphPad Software Inc., San Diego, CA, USA) was used for all analyses.

## Figures and Tables

**Figure 1 ijms-23-10247-f001:**
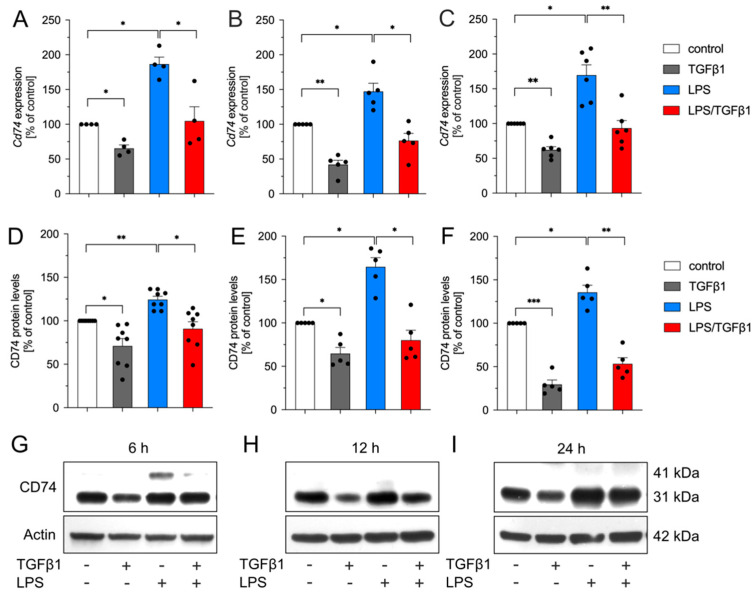
TGFβ1 inhibits LPS-mediated upregulation of Cd74 in BV2 cells. Expression of *Cd74* after treatment of BV2 cells with TGFβ1 (5 ng/mL), LPS (1 µg/mL), or both factors for 6 h (**A**), 12 h (**B**), and 24 h (**C**). Quantification of CD74 total protein levels after treatments for 6 h (**D**), 12 h (**E**), and 24 h (**F**). (**G**–**I**) Representative western blot images for each experimental time point. Data are given as means ± SEM for at least four independent experiments. *p*-values derived from one-way ANOVA followed by Tukey’s multiple comparison tests are shown as follows: * *p* < 0.05, ** *p* < 0.01, and *** *p* < 0.001.

**Figure 2 ijms-23-10247-f002:**
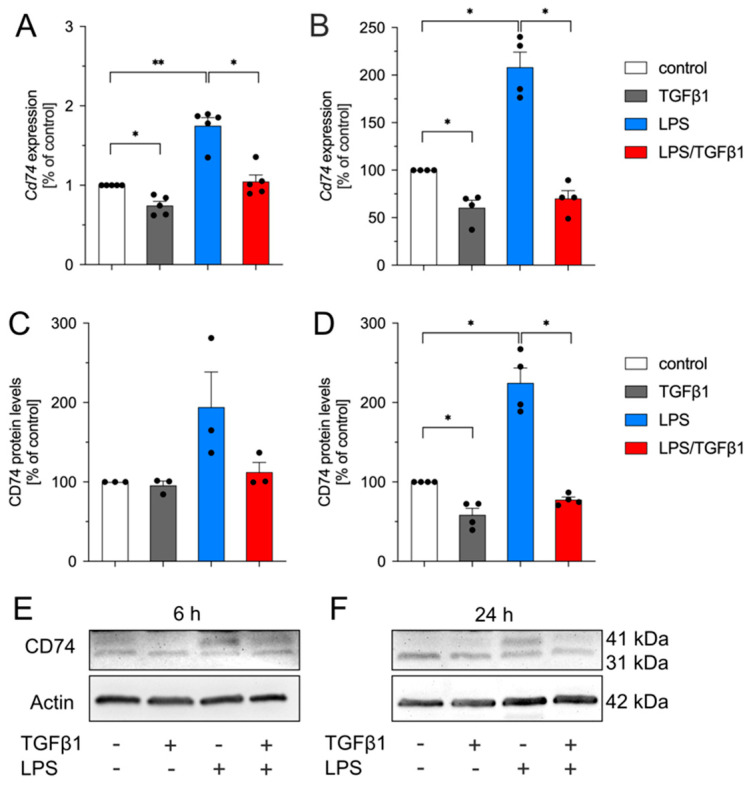
TGFβ1 inhibits LPS-induced expression of *Cd74* in primary microglia. Expression of *Cd74* after treatment of primary microglia with TGFβ1 (5 ng/mL), LPS (1 µg/mL), or a combination of both factors for 6 h (**A**) and 24 h (**B**). Quantifications of CD74 total protein levels after treatments for 6 h (**C**) and 24 h (**D**). (**E**,**F**) Representative western blot images for the analyzed experimental time points. Data are given as means ± SEM for at least three independent experiments. *p*-values derived from one-way ANOVA followed by Tukey’s multiple comparison tests are shown as follows: * *p* < 0.05 and ** *p* < 0.01.

**Figure 3 ijms-23-10247-f003:**
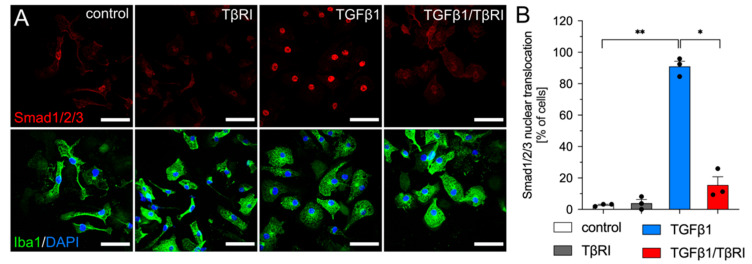
TβRI efficiently blocks TGFβ1-induced SMAD nuclear translocation in primary microglia. Primary microglia were treated with TβRI (500 nM), TGFβ1 (5 ng/mL), or a combination of both reagents for 2 h under serum-free conditions. (**A**) Representative immunocytochemistry images after staining with anti-Iba1 and anti-SMAD1/2/3 primary antibodies and fluorescence-coupled secondary antibodies. DAPI was used to counterstain nuclei. (**B**) Quantifications of microglia with nuclear SMAD accumulation. Scale bars indicate 50 µm. Data are given as means ± SEM for three independent experiments. *p*-values derived from one-way ANOVA followed by Tukey’s multiple comparison tests are shown as follows: * *p* < 0.05 and ** *p* < 0.01.

**Figure 4 ijms-23-10247-f004:**
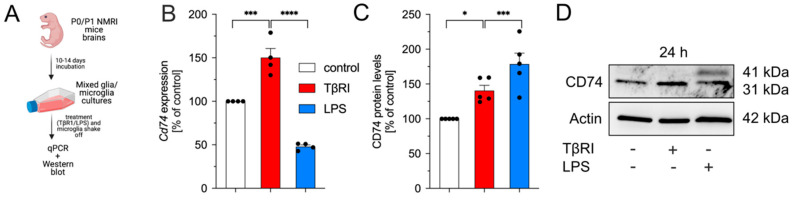
Inhibition of TGFβ signaling increases expression of *Cd74* in mixed glia cultures. (**A**) Scheme of experimental workflow, created with BioRender.com. (**B**) Expression of *Cd74* after treatment of mixed glia cultures with TβRI (500 nM) or LPS (1 µg/mL) for 24 h. (**C**) Quantifications of CD74 total protein levels after treatments for 24 h. (**D**) Representative western blot images for the analyzed experimental time point. Data are given as means ± SEM for at least four independent experiments. *p*-values derived from one-way ANOVA followed by Tukey’s multiple comparison tests are shown as follows: * *p* < 0.05, *** *p* < 0.001, and **** *p* < 0.0001.

**Figure 5 ijms-23-10247-f005:**
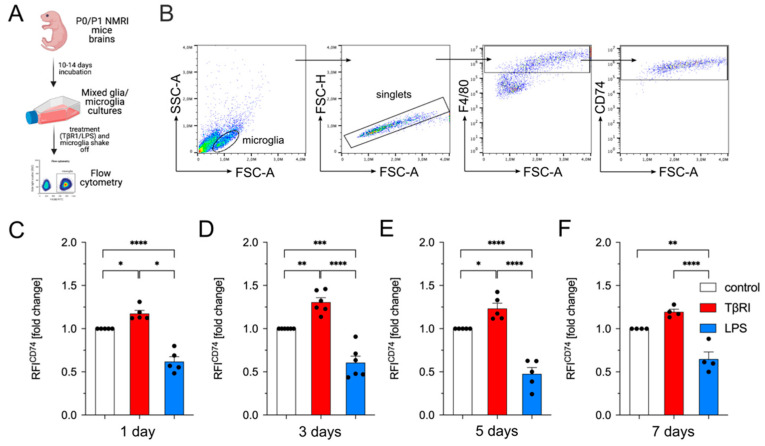
Inhibition of TGFβ signaling increases surface expression of CD74 in microglia. (**A**) Scheme of experimental workflow for flow cytometry, created with BioRender.com. (**B**) Gating strategy to detect CD74 surface expression in F4/80^+^ microglia. Relative fluorescence intensities (RFI) of CD74 in microglia (out of F4/80^+^ cells) were analyzed after 1 (**C**), 3 (**D**), 5 (**E**), and 7 days (**F**). (**C**–**F**) Quantifications and statistical analyzes of relative fluorescence intensities (RFI) of CD74 after indicated treatments and time points. Data are given as means ± SEM for at least three independent experiments. *p*-values derived from one-way ANOVA followed by Tukey’s multiple comparison tests are shown as follows: * *p* < 0.05, ** *p* < 0.01, *** *p* < 0.001, and **** *p* < 0.0001.

**Figure 6 ijms-23-10247-f006:**
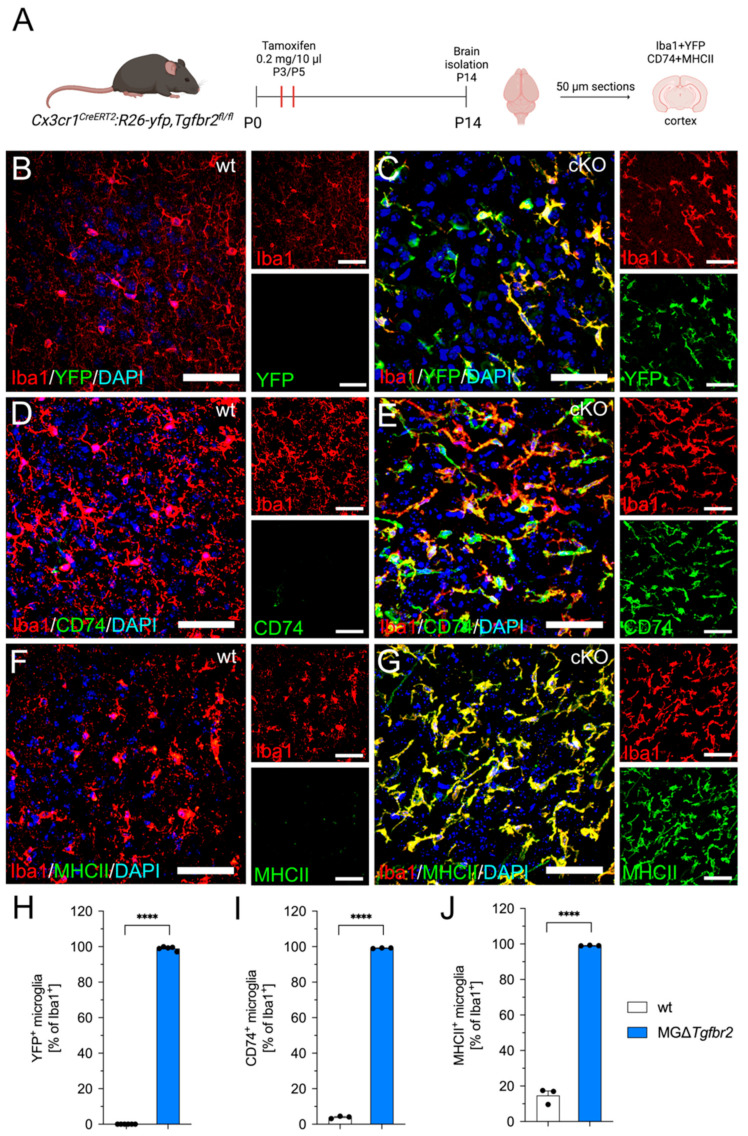
Loss of microglial *Tgfbr2* expression results in upregulation of CD74 in vivo. (**A**) Scheme of the experimental workflow for analysis of YFP, CD74, and MHCII expression in frontal cortices of microglia-specific *Tgfbr2* mutant mice. Created with BioRender.com. (**B**,**C**) Immunohistochemistry against for Iba1 (microglia) and YFP (reporter gene) to detect and quantify recombination efficacy. Double staining for Iba1 and CD74 to detect and quantify CD74^+^ microglia in control (**D**) and knockout mice (**E**). Immunohistochemical detection of MHCII and Iba1 in control mice (**F**) and *Tgfbr2*-deficient mice (**G**). Representative images at a magnification of 60× are shown. Scale bars indicate 50 µm. Quantifications of YFP^+^ (**H**), CD74^+^ (**I**), and MHCII^+^ (**J**) microglia in control and mutant mice. Data are given as means ± SEM for at least three mice per group. *p*-values derived from student’s *t*-test are shown as follows: **** *p* < 0.0001.

## Data Availability

The data that support the findings of this study are available from the corresponding author upon reasonable request.

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
