# Peer review of "Microglial CD74 Expression Is Regulated by TGFβ Signaling"

_ijms, 2022, doi:10.3390/ijms231810247_

Round 1

Reviewer 1 Report

Jannik Jahn et al. have submitted a manuscript entitled "Microglial CD74 expression is regulated by TGFb signaling." The authors try to show that CD74 is regulated downstream of the TGFb signaling in different microglia cell models and animal model. And CD74 can be a marker of microglia activation. To my knowledge, this is the first time showing CD74 is downstream of TGFb. The authors may need to address several major concerns before this manuscript can be published. 

Major Concerns:

1. For the primary microglia culture experiments, as the authors used the "shaking" method to get rid of astrocytes, it would be better to provide data to indicate the purity and verify the identity of primary microglia for the downstream analysis. 

2. Did LPS incubation also affect SMAD translocation? In addition, the authors are trying to demonstrate that TGFb1-SMAD signaling suppresses Cd74 expression, it would also be important to show knocking down the SMAD1/2/3 or blockage of SMAD1/2/3 nuclear translocation under TGFb1 stimulation could remove the inhibitory effects on Cd74 expression. 

3. Apart from studying the nuclear translocation of SMAD1/2/3, the active phosphorylation level of SMAD should also be examined in Fig 3 experimental groups.

Minor Concern:

1. In lines 166-168, the authors mention "We observed that inhibition 166 of TGFβ signaling in mixed glia cultures resulted in increased surface levels of CD74 in 167 microglia at all time points analyzed (Fig. 5C-F)" Is the x-axis of Fig 5c-f trimmed? Please annotate the time-point for each figure.

2. The microglia-specific Tgfbr2 knockout mice have been used in this study, please describe in the methodology. 

3. Please check if Fig 4b-d are clearly described while the authors have mentioned. Is Fig 4b a RNA work as the y-axis annotates Cd74 instead of CD74. Line 148 mentions it is a protein. Which bar chart belongs to Figure 4D. The legend of Fig 4c should be an analysis instead of western blot image.

Author Response

Response to reviewer comments

Reviewer 1:

Major Concerns:

  1. For the primary microglia culture experiments, as the authors used the "shaking" method to get rid of astrocytes, it would be better to provide data to indicate the purity and verify the identity of primary microglia for the downstream analysis. 

Response: The authors acknowledge this comment It is crucial that the primary microglia cultures show high purity. As described in several previous publications [1–4], we generate microglia cultures with a purity of >95 % as reflected by stainings with FITC-coupled tomatolection or isolectin and/or Iba1 immunocytochemistry. Figure 3 shows the purity of cultures used in the present study as demonstrated by Iba1 immunolabeling.

  1. Did LPS incubation also affect SMAD translocation? In addition, the authors are trying to demonstrate that TGFb1-SMAD signaling suppresses Cd74 expression, it would also be important to show knocking down the SMAD1/2/3 or blockage of SMAD1/2/3 nuclear translocation under TGFb1 stimulation could remove the inhibitory effects on Cd74 expression. 

Response: This is a great comment. In a different project were are currently addressing how LPS (and other M1-like triggers) interfere with TGFb signaling. At least for LPS, a negative feedback blocking TGFb signaling and subsequent Smad phosphorylation induced by LPS-driven upregulation of the inhibitory Smad7 could be observed. However, this is part of separate study which will (hopefully) be finished quite soon. Thus, we have not addressed this issue in the present study.

  1. Apart from studying the nuclear translocation of SMAD1/2/3, the active phosphorylation level of SMAD should also be examined in Fig 3 experimental groups.

Response: We agree that the phosphorylation levels of R-Smads are essential for TGFb downstream signalling. However, nuclear translocation of Smads only takes place after receptor-mediated phosphorylation of Smad2/3. Moreover, we have demonstrated that the translocation of normally phosphorylated Smad2 can be significantly impaired in the presence of high calcium concentrations [5]. Thus, nuclear translocation of Smads is a readout beyond phosphorylation and even more suitable to prove the functionality of the inhibitor used in this study.

Minor Concern:

  1. In lines 166-168, the authors mention "We observed that inhibition 166 of TGFβ signaling in mixed glia cultures resulted in increased surface levels of CD74 in 167 microglia at all time points analyzed (Fig. 5C-F)" Is the x-axis of Fig 5c-f trimmed? Please annotate the time-point for each figure.

Response: The authors acknowledge this comment. We have placed the legend for the x-axis in Figure 5F. The time-points for each figure has been added accordingly. Again, thank you for this valuable comment.

  1. The microglia-specific Tgfbr2 knockout mice have been used in this study, please describe in the methodology. 

Response: The authors thank the reviewer for this crucial comment. The details covering generation of microglia-specific Tgfbr2 knockout mice have been added to the methods part of the manuscript.

  1. Please check if Fig 4b-d are clearly described while the authors have mentioned. Is Fig 4b a RNA work as the y-axis annotatesCd74instead of CD74. Line 148 mentions it is a protein. Which bar chart belongs to Figure 4D. The legend of Fig 4c should be an analysis instead of western blot image.

Response: Thank for the comment. We have checked Fig.4 B-D accordingly. Indeed, Fig.4B refers to RNA – as the y-axis annotation indicates – and Fig. 4C depicts protein levels. We have slightly modified the text of the corresponding results section to make this point clearer.

References

  1. Attaai, A.; Neidert, N.; von Ehr, A.; Potru, P.S.; Zöller, T.; Spittau, B. Postnatal Maturation of Microglia Is Associated with Alternative Activation and Activated TGFβ Signaling. Glia 2018, 66, 1695–1708, doi:10.1002/glia.23332.
  2. Neidert, N.; von Ehr, A.; Zöller, T.; Spittau, B. Microglia-Specific Expression of Olfml3 Is Directly Regulated by Transforming Growth Factor Β1-Induced Smad2 Signaling. Front Immunol 2018, 9, 1728, doi:10.3389/fimmu.2018.01728.
  3. von Ehr, A.; Attaai, A.; Neidert, N.; Potru, P.S.; Ruß, T.; Zöller, T.; Spittau, B. Inhibition of Microglial TGFβ Signaling Increases Expression of Mrc1. Front Cell Neurosci 2020, 14, 66, doi:10.3389/fncel.2020.00066.
  4. Spittau, B.; Wullkopf, L.; Zhou, X.; Rilka, J.; Pfeifer, D.; Krieglstein, K. Endogenous Transforming Growth Factor-Beta Promotes Quiescence of Primary Microglia in Vitro. Glia 2013, 61, 287–300, doi:10.1002/glia.22435.
  5. Ming, M.; Manzini, I.; Le, W.; Krieglstein, K.; Spittau, B. Thapsigargin-Induced Ca(2+) Increase Inhibits TGFbeta1-Mediated Smad2 Transcriptional Responses via Ca(2+)/Calmodulin-Dependent Protein Kinase II. J Cell Biochem2010, doi:10.1002/jcb.22843.
  6. Zöller, T.; Schneider, A.; Kleimeyer, C.; Masuda, T.; Potru, P.S.; Pfeifer, D.; Blank, T.; Prinz, M.; Spittau, B. Silencing of TGFβ Signalling in Microglia Results in Impaired Homeostasis. Nat Commun 2018, 9, 4011, doi:10.1038/s41467-018-06224-y.

Reviewer 2 Report

This paper by Jahn et al. presents solid work demonstrating CD74 expression is regulated by TGFbeta signaling. The presented data are very convincing and the authors deserve appreciations for (1) presenting all original blots with sufficient experimental repeats; (2) having diagrams explaining experimental procedures along the figures. I just have a few comments:
1. The methods related to Figure 6, especially animal use need to be added.
2. The 41 kDa & 31 kDa isoforms were observed in most figures and their expression across time points and treatments show different dynamics. I suggested adding panels or having supplementary figures to present specifically the protein levels of those isoforms instead of only presenting the aggregated "CD74 protein levels".
3. The 41 kDa isoform was upregulated in a faster manner than the 31 kDa isoform in BV2 cells and to a lesser degree in primary microglia cells. I would suggest discussing along with the intriguing result where 31 kDa was upregulated by TGFb inhibition.
4. The qPCR result can be further improved if the authors probe the isoform expression by using primer pairs that specifically identify particular isoforms. If the data are not available, please explicitly explain the isoform(s) your primer is targeting. And it will be nice to include in the discussion.
5. CD74 is a glycosylated protein but I did not see any method description in chemical deglycosylation but the blot presents no smearing pattern suggesting no glycosylation. Could you please explain?
6. In Figure 6H-J, I would suggest that the authors also include data with CD74+/MHCII+ microglial cells that are either YFP+ or YFP- (rather than just summarized data). It may shed light on possibly bystander effects as even those Tgfbr2-intact microglial cells (YFP-) may display weaker activation/antigen presentation. Further, TAM was treated almost 10 days before the cell isolation ruling out the possibility that the YFP is still translating. It may imply that the TGFb signaling may be an autocrine signaling pathway within microglial cells considering microglial cells as the source. Or the weak activation endorsed by TGFb signal altered the microenvironment results in weaker activation even if TGFb signaling remains intact.
7. It would be also worthwhile to discuss the source of TGFb in the discussion.

Author Response

Reviewer 2:

This paper by Jahn et al. presents solid work demonstrating CD74 expression is regulated by TGFbeta signaling. The presented data are very convincing and the authors deserve appreciations for (1) presenting all original blots with sufficient experimental repeats; (2) having diagrams explaining experimental procedures along the figures. I just have a few comments: 
1. The methods related to Figure 6, especially animal use need to be added. 

Response: The authors thank the reviewer for this crucial comment. The details covering generation of microglia-specific Tgfbr2 knockout mice have been added to the methods part of the manuscript.

  1. The 41 kDa & 31 kDa isoforms were observed in most figures and their expression across time points and treatments show different dynamics. I suggested adding panels or having supplementary figures to present specifically the protein levels of those isoforms instead of only presenting the aggregated "CD74 protein levels". 

Response: We are grateful for this valuable comment. We have added a supplementary file showing the densitometric evalutions of 31 kDa and 41 kDa in primary microglia. The results part has been modified accordingly.

  1. The 41 kDa isoform was upregulated in a faster manner than the 31 kDa isoform in BV2 cells and to a lesser degree in primary microglia cells. I would suggest discussing along with the intriguing result where 31 kDa was upregulated by TGFb inhibition. 

Response: Thank you very much. We have added a section in the discucssion covering this interesting issue.

  1. The qPCR result can be further improved if the authors probe the isoform expression by using primer pairs that specifically identify particular isoforms. If the data are not available, please explicitly explain the isoform(s) your primer is targeting. And it will be nice to include in the discussion. 

Response: The primer pair to detect Cd74 used throughout the study, bind at bp 177-195 (forward primer) and bp 260-240 (reverse primer) of the more than 1200 bp sized mRNA which reflects exon 2. The difference between the distinct isoforms of Cd74 lies within the c-terminus of the protein. Thus the Cd74 qPCR primers detect both Cd74 isoforms.

  1. CD74 is a glycosylated protein but I did not see any method description in chemical deglycosylation but the blot presents no smearing pattern suggesting no glycosylation. Could you please explain? 

Response: Throughout the protein isolation, SDS-page and western blot procedure, we have not used any chemical deglycosylation steps. In all blots, we never observed the typical smear of glycosylated proteins. We assume that our reducing conditions and the protein boiling have resulted in deglycosylation of CD74.

  1. In Figure 6H-J, I would suggest that the authors also include data with CD74+/MHCII+ microglial cells that are either YFP+ or YFP- (rather than just summarized data). It may shed light on possibly bystander effects as even those Tgfbr2-intact microglial cells (YFP-) may display weaker activation/antigen presentation. Further, TAM was treated almost 10 days before the cell isolation ruling out the possibility that the YFP is still translating. It may imply that the TGFb signaling may be an autocrine signaling pathway within microglial cells considering microglial cells as the source. Or the weak activation endorsed by TGFb signal altered the microenvironment results in weaker activation even if TGFb signaling remains intact. 

Response: Thank you very much for this valuable input. Basically, we agree that this discrimination between YFP+ and YFP- cells is important. However, we have a recombination rate of 99% and it is hard find any YFP- negative microglia in these brain sections. Additionally, we observed 99% CD74+ and 99% MHCII+ microglia in our study which indicated that YFP+ microglia are positive for both markers, CD74 as well as MHCII.

After TAM application, the Cre activates YFP expression which is stable until the affected microglia cells is alive. The same is true for our gene of interest (Tgfbr2) which is permanently deleted in recombined microglia. Based on this, and the previous study characterizing this mutant strain [6], we are sure that we have microglia with deficiency of TGFb signaling 10 days after recombination.

The source of TGFb is a very interesting aspect. We have demonstrated that especially neurons are expressing TGFb1 and to a much lesser extent also astrocytes, oligodendrocytes, and microglia [1]. However, activated microglia have been shown to increase expression of TGFb1 [4]. In this context, there could be the try to rescue microglia from the activation phenotype by releasing TGFb1 which should act in a autocrine manner. The paracrine (neuroprotective) effects of microglia-released TGFb1 on neuron survival could be a very important in vivo mechanism in our mutant mice. However, these mechanisms are currently under investigation in a distinct project.

  1. It would be also worthwhile to discuss the source of TGFb in the discussion.

Response: Indeed, it is worthwhile. We have included this issue in the discussion section of the manuscript. Again, thanks for this comment.

References

  1. Attaai, A.; Neidert, N.; von Ehr, A.; Potru, P.S.; Zöller, T.; Spittau, B. Postnatal Maturation of Microglia Is Associated with Alternative Activation and Activated TGFβ Signaling. Glia 2018, 66, 1695–1708, doi:10.1002/glia.23332.
  2. Neidert, N.; von Ehr, A.; Zöller, T.; Spittau, B. Microglia-Specific Expression of Olfml3 Is Directly Regulated by Transforming Growth Factor Β1-Induced Smad2 Signaling. Front Immunol 2018, 9, 1728, doi:10.3389/fimmu.2018.01728.
  3. von Ehr, A.; Attaai, A.; Neidert, N.; Potru, P.S.; Ruß, T.; Zöller, T.; Spittau, B. Inhibition of Microglial TGFβ Signaling Increases Expression of Mrc1. Front Cell Neurosci 2020, 14, 66, doi:10.3389/fncel.2020.00066.
  4. Spittau, B.; Wullkopf, L.; Zhou, X.; Rilka, J.; Pfeifer, D.; Krieglstein, K. Endogenous Transforming Growth Factor-Beta Promotes Quiescence of Primary Microglia in Vitro. Glia 2013, 61, 287–300, doi:10.1002/glia.22435.
  5. Ming, M.; Manzini, I.; Le, W.; Krieglstein, K.; Spittau, B. Thapsigargin-Induced Ca(2+) Increase Inhibits TGFbeta1-Mediated Smad2 Transcriptional Responses via Ca(2+)/Calmodulin-Dependent Protein Kinase II. J Cell Biochem2010, doi:10.1002/jcb.22843.
  6. Zöller, T.; Schneider, A.; Kleimeyer, C.; Masuda, T.; Potru, P.S.; Pfeifer, D.; Blank, T.; Prinz, M.; Spittau, B. Silencing of TGFβ Signalling in Microglia Results in Impaired Homeostasis. Nat Commun 2018, 9, 4011, doi:10.1038/s41467-018-06224-y.

Reviewer 3 Report

This is a very sound paper that presents in a very concisely and structured way the data. The message is very clear.

Author Response

This is a very sound paper that presents in a very concisely and structured way the data. The message is very clear.

Response: The authors thank the reviewer for her/his comments.